# Conduct Disorder in Immigrant Children and Adolescents: A Nationwide Cohort Study in Sweden

**DOI:** 10.3390/ijerph182010643

**Published:** 2021-10-11

**Authors:** Mehdi Osooli, Henrik Ohlsson, Jan Sundquist, Kristina Sundquist

**Affiliations:** 1Center for Primary Health Care Research, Lund University, SE-221 00 Malmö, Sweden; henrik.ohlsson@med.lu.se (H.O.); jan.sundquist@med.lu.se (J.S.); kristina.sundquist@med.lu.se (K.S.); 2Department of Family Medicine and Community Health, Icahn School of Medicine at Mount Sinai, New York, NY 10029, USA; 3Department of Population Health Science and Policy, Icahn School of Medicine at Mount Sinai, New York, NY 10029, USA; 4Center for Community-Based Healthcare Research and Education (CoHRE), Department of Functional Pathology, School of Medicine, Shimane University, Shimane 690-8504, Japan

**Keywords:** adolescents, children, conduct disorder, cohort, real-world data, register-based research, immigrants, Sweden

## Abstract

Introduction. Conduct disorder is a psychiatric diagnosis characterized by repetitive and persistent norm-breaking behavior. This study aimed to compare the risk of conduct disorder between first- and second-generation immigrant children and adolescents and their native controls. Methods. In this nationwide, open-cohort study from Sweden, participants were born 1987–2010, aged 4–16 years at baseline, and were living in the country for at least one year during the follow-up period between 2001 and 2015. The sample included 1,902,526 and 805,450 children-adolescents with native and immigrant backgrounds, respectively. Data on the conduct disorder diagnoses were retrieved through the National Patient Register. We estimated the incidence of conduct disorder and calculated adjusted Hazard Ratios. Results. Overall, the adjusted risk of conduct disorder was lower among first-generation immigrants and most second-generation immigrant groups compared with natives (both males and females). However, second-generation immigrants with a Swedish-born mother and a foreign-born father had a higher risk of conduct disorder than natives. Similar results were found for sub-diagnoses of conduct disorder. Conclusions. The higher risk of conduct disorder among second-generation immigrants with a Swedish-born mother and the lower risk among most of the other immigrant groups warrants special attention and an investigation of potential underlying mechanisms.

## 1. Introduction

According to United Nation (UN) estimates from the last decade, an increasing number of people have left their home countries to seek a safer and better life elsewhere [1]. War, conflict, and political and economic instability have jointly driven the current immigration crisis. As a result, many countries worldwide may receive even more immigrants in the coming years.

Immigrants experience substantial stress before, during, and after immigration. Upon arriving in the host country, issues like struggles with adapting to a new culture and securing jobs and housing can affect immigrants’ own as well as their children’s mental health [2]. Missing their country of origin and the loved ones left behind may additionally pose a high psychological burden on immigrants. Immigrant children and adolescents might have a higher risk of mental health issues because of their dependence on parents and impartial coping mechanisms. However, despite the importance of examining immigrant mental health, high-quality studies in this area are limited, and previous research on young immigrants’ mental health have shown inconsistent results. In a survey from The Netherlands, first- and second-generation Moroccan adolescents had lower self-reported externalizing behavior problems compared with their native peers [3]. However, an increasing trend of both internalizing and externalizing disorders over time was observed among immigrants, suggesting their diminishing mental health over time. A European study reported lower rates of externalizing disorders among 14–15-year-old immigrants compared with natives [4] and studies from the USA and Canada have shown a lower prevalence of externalizing problem behaviors among immigrants compared with natives [5]. Recent systematic reviews have reported that most studies on immigrant mental health had suboptimal designs such as low sample sizes and problems with bias [6,7,8].

Conduct disorder is a psychiatric disorder in children and adolescents that is characterized by repetitive and persistent norm-breaking behavior and violation of other people’s rights [9]. Affecting approximately 2% of children and adolescents globally [10], the disorder is responsible for 1% of the total years lived with disability [11]. Some data suggest an increasing trend of conduct disorder and other externalizing disorders over recent years [12,13]. A mix of genetic, family-related, environmental factors, and their interplay might explain the development of conduct disorder [9,14,15]. In addition, adolescents with conduct problems are more likely to commit crimes, get sexually abused, or use drugs [16,17], which may further deteriorate their mental health. Poor educational attainment and career failure may negatively affect the health-related quality of life of these individuals in adulthood [18,19]. Despite the negative effects of conduct disorder, little is known about the incidence across settings and subpopulations [9], and more knowledge of this disorder is particularly important in immigrants as the many negative effects of this disorder could hinder integration in the new society. It is also possible that the psychosocial stress related to migration may influence children’s and adolescents’ mental health. However, data from children and adolescents with an immigrant background is particularly limited [20,21,22]. This is partly due to limited availability of nationwide datasets that contain relevant determinants of health-related outcomes.

Sweden has received many refugees and non-refugee immigrants over the past few decades. By 2018, approximately one-third of the Swedish population were first- or second-generation immigrants [23]. In the 5–24 year age group, approximately 26% are first- or second-generation immigrants [24]. Sweden has also excellent register resources with national coverage. These registers have individual-level longitudinal data on various social, welfare, and health outcomes, which make them suitable for studies on immigrants’ mental health. In our recent studies, we have shown that the incidence of depression [25] and Attention Deficit Hyperactivity Disorders (ADHD) [26] varies across immigrant groups and in comparison with the native population in Sweden. However, longitudinal studies on conduct disorders in immigrants are still limited worldwide. This study therefore aimed to examine the incidence of conduct disorder in first- and second-generation immigrant children and adolescents compared with natives in Sweden, taking socioeconomic factors into account. Identifying subpopulations at higher risks of conduct disorder is a crucial step in planning and delivering prevention and treatment interventions.

## 2. Materials and Methods

This was a population-based retrospective open cohort study. Pseudonymized data from several nationwide registers were provided to us by Statistics Sweden (SCB) [20] and The National Board of Health and Welfare (In Swedish: Socialstyrelsen) [27] for this study. A unique serial number was used to link data across registers at the individual level. All study procedures complied with the ethical principles in the Declaration of Helsinki [28]. This study was part of a larger project that received ethical approval from the Regional Ethical Review Board in Lund, Sweden (Ethics approval No: 2012/795). The latest amendment was approved by the Swedish Ethical Review Authority 12 March 2019 (Dnr: 2019-01588)”.

### 2.1. Participants and Observation

Eligible participants were born 1987–2010, and were registered as residents of Sweden during at least one year during the study period between 1 January 2001 and 31 December 2015. The enrollment included individuals aged 4–16 years at baseline, which could occur between 1 January 2001 and 31 December 2013. In total 2,707,976 children-adolescents were included in the study population that consisted of 1,902,526 (70.3%) natives (defined as Swedish-born with both parents born in Sweden), 228,889 (8.4%) first-generation immigrants (defined as born abroad with both parents born abroad), 160,839 (5.9%) second-generation immigrants with a foreign-born father (and a native mother), 133,217 (4.9%) second-generation immigrants with a foreign-born mother (and a native father), and 282,505 (10.4%) second-generation immigrants with two foreign-born parents. All second-generation immigrant groups were defined as born in Sweden.

The follow-up continued until participants turned 18 years, emigrated, died, or at the end of the study on 31 December 2015, whichever came first. Approximately 90% of the participants from each of the native/immigrant groups were followed until the end of the study or the age of 18 years. In total, we accrued a total of 20,181,775 person-years of observations from participants including the following: natives (14,741,074 person-years), first-generation immigrants with two foreign-born parents (1,139,125), second-generation immigrants with two foreign-born parents (2,103,371), second-generation immigrants with a foreign-born mother (991,173), and second-generation immigrants with a foreign-born father (1,207,032).

### 2.2. Registers and Data

We used the following nationwide registers: The Register of the Total Population (RTB), the Multi-Generation Register, the Medical Birth Register, the Cause of Death register, the Migration Register, the Longitudinal Integration Database for Health Insurance and Labor Market Studies (LISA), and The National Patient Register (NPR). Data on conduct disorder diagnoses were obtained from the National Patient Register (NPR). The NPR had full national coverage for inpatient and outpatient hospitalization records during the entire study period and has been validated for research purposes [29].

### 2.3. Maternal Income

We used maternal income as a surrogate of socioeconomic status. We obtained data on maternal income from the LISA database at Statistics Sweden. Statistics Sweden calculates disposable income based on total income from all potential sources, including wages, welfare benefits, other social subsidies, and pensions. We obtained annual disposable income for all females 18–65 years living in Sweden during the observation period. To account for inflation, we categorized income into quintiles relative to all other females assigned a disposable income in that year. We then created year-specific income quintiles and used the maternal income quintile for the year of start of follow-up for each participant. Few individuals lacked information on income in the total study population. As a higher proportion of foreign-born parents lacked information on income, we did not impute missing data on income since we judged that this information was not missing at random between subgroups. However, we included those with missing income values in the lowest income quintile.

### 2.4. Conduct Disorder Diagnosis

In the NPR all psychiatric diagnoses for the study period were registered using the 10th version of the International Classification of Disease (ICD 10) [30]. We searched for outpatient or inpatient hospital diagnoses with the three-digit code F91 to identify participants with a conduct disorder diagnosis. We also extracted data on specific subtypes of conduct disorder using the following four-digit ICD codes: confined to family context (F91.0), CD with aggression and lack of social adjustment (F91.1), CD with aggression but normal social adjustment (F91.2), oppositional defiant disorder (ODD) (F91.3) and other and unspecified (F91.8, F91.9).

### 2.5. Statistical Analysis

We estimated non-adjusted incidence rates of conduct disorder per 100,000 person-years and report 95% confidence intervals (CI) for the estimates. Using Cox regression, we ran sex-specific models and estimated hazard ratios (HRs) of conduct disorder with 95% CI for the immigrant groups compared with natives in two steps. In Model 1, we controlled for birth year and age at the start of the follow up. In Model 2, we additionally controlled for maternal income. Maternal income (see below) could not be attained for some participants including some of those from the first-generation immigrant group (*n* = 23,392; 10.2%); a majority of these did not have their biological mother in Sweden (*n* = 16,955; 7.4%). We assigned the lowest maternal income value (first income quintile) to those with a missing maternal income. We evaluated the proportional hazards assumptions for the Cox regression models using log-log survival plots and did not find any meaningful deviations from the assumption indicating that there was no effect modification from time in the models. We used Stata (version 15.1) for data management and statistical analysis [31].

## 3. Results

Table 1 shows the characteristics of the study participants aged 4–16 years at baseline; i.e., natives, first-generation immigrants, and second-generation immigrants, the latter further divided into the three subgroups described above. Approximately one-quarter of natives (27.4%) compared with approximately three-quarters of the first-generation immigrants (77.3%) belonged to the lowest maternal income quintile. Among the second-generation immigrants, the lowest proportion of individuals in the lowest maternal income quintile was among those with a foreign-born mother and Swedish-born father (31.7%) and the highest proportion of individuals in the lowest maternal income quintile was among those with both parents born abroad (64.8%). In total, 9088 individuals, including 6440 natives, 454 first-generation immigrants, 780 s-generation immigrants with two foreign-born parents, 473 s-generation immigrants with a foreign-born mother, and 941 s-generation immigrants with a foreign-born father, had a registered conduct disorder diagnosis in the NPR.

Table 2 shows that there were more male than female participants with a conduct disorder diagnosis (any subtype). Around two thirds (6203) were male. Table 2 also shows the non-adjusted incidence rates of conduct disorder among male and female study participants in the different groups. The incidence rates of conduct disorder ranged from 51.0 to 104.6 among males and 23.8 to 51.5 among females. In both males and females, the highest and lowest incidence rates of conduct disorder were found among the second-generation immigrants with a foreign-born father and the second-generation immigrants with two foreign-born parents, respectively. Across native and immigrant subgroups, the incidence rate of conduct disorder was approximately twice as high in males compared with females.

Table 3 shows the HRs of conduct disorder among the male and female immigrant groups compared with natives. In both males and females, first-generation immigrants and second-generation immigrants with two foreign-born parents had significantly lower hazards of conduct disorder compared with natives in both Model 1 (adjusted for birth year and age at baseline) and Model 2 (also adjusted for maternal income). In second-generation immigrants with only one foreign-born parent a different pattern emerged. Males and females with a foreign-born father and Swedish-born mother had the highest hazards of conduct disorder. Their significant HRs in Model 1 were 1.79 and 1.82, respectively. In Model 2, these risks only decreased slightly to 1.71 and 1.73, respectively, and remained significant. Males with a foreign-born mother and a Swedish-born father had a slightly but statistically non-significant higher risk of conduct disorder compared with natives; in Model 2, the HR was 1.04 (95% CI = 0.93, 1.16). The corresponding HR for females was 1.18 (95% CI = 1.00, 1.39).

In supplementary analyses of sub diagnoses, most of the cases were found to belong to ODD and other and unspecified conduct disorders in both males and females (Appendix A
Table A1). With few exceptions, most results for the sub diagnoses were consistent with the overall results. Both male and female second-generation immigrants with a foreign-born father and a Swedish-born mother had higher HRs of all subtypes of conduct disorder and almost all of these were statistically significant (Table A2 and Table A3).

## 4. Discussion

Using nationwide registers, we conducted a longitudinal study to compare the incidence of conduct disorder between first- and second-generation immigrant children and adolescents and their native controls in Sweden. Our results indicated a markedly higher risk of conduct disorder among male and female second-generation immigrants with a foreign-born father and a Swedish-born mother whereas males and females with a foreign-born mother and a Swedish-born father had slightly higher risks that where non-significant among the males. In contrast, in both males and females, first-generation immigrants and second-generation immigrants with two foreign-born parents had a lower risk of conduct disorder.

Our study indicated that a potential mental health advantage in first-generation immigrants seems to partly decrease in the second-generation immigrant children and adolescents. Our results are somewhat but not entirely in line with findings from studies across Europe and the USA as few studies have focused on conduct disorder, which belongs to the category externalizing disorders where ADHD also is included [3,4,5]. A recent systematic review of the prevalence of psychiatric disorders in child and adolescent refugees and asylum seekers only found two studies for the conduct disorder subtype ODD with an estimated average prevalence of 1.69% (95% CI: 0.78, 4.16) for the condition in this group [8]. Our finding on markedly higher risks of conduct disorder in male and female second-generation immigrants with a foreign-born father and a Swedish-born mother was in line with results from a study from Finland. The Finnish study found higher odds of externalizing disorders among offspring of foreign-born fathers [32]. However, the Finnish study also found higher odds of ADHD in the offspring of two foreign-born parents, which is in contrast with our results focusing on conduct disorders.

We are unaware of previous studies focusing on subtypes of conduct disorders, possibly due to small sample sizes. Although most of our findings for the subtypes were consistent with the overall results, these results are unique and could be helpful in future studies on potential mechanisms behind our findings.

Besides genetic factors, a set of prenatal (e.g., maternal stress, smoking, and drug use), perinatal (e.g., parental psychopathology and malnutrition), familial (e.g., maladaptive and harsh parenting), and neighborhood risk factors (e.g., deviant peers and poverty) may contribute to the risk of developing conduct disorder [9]. Despite its high burden, even in high-income countries such as the UK and the USA, research on conduct disorder has received substantially less funding compared with many other psychiatric diseases [9]. Our knowledge about causal pathways and the weight of various risk factors in the development of conduct disorder is therefore still at very early stages. This study did, however, not intend to examine the causal effects of immigration on risk of conduct disorder although we attempt to discuss potential explanations behind our findings.

Among all immigrant groups, only those second-generation immigrants with a foreign-born father and a Swedish-born mother had markedly higher risks of conduct disorder compared with natives while first-generation immigrant children and adolescents as well as second-generation immigrants with two foreign-born parents had lower risks. All residents in Sweden are entitled to free healthcare until they turn 18 years old. Moreover, schools are actively engaged in maintaining the mental health of their students, but the use of these services may be hampered because some immigrant populations have different patterns of using healthcare services for their children’s psychiatric problems [33,34]. Poor health literacy, cultural barriers, and stigma related to psychiatric disorders are more common among immigrant parents than natives and especially in those with a low socioeconomic status [35,36]. On the other hand, because conduct disorder involves the violation of other peoples’ rights, lower service utilization may play a less important role in its diagnosis compared with other psychiatric disorders. Raising awareness on early symptoms of conduct disorder among parents, guardians, and teachers can help to ensure early diagnosis and start of psychosocial support among affected children and adolescents.

Most known non-genetic risk factors of conduct disorder are caused or associated with parental stress and conflict in the family. Marriage or partnership between a native and an immigrant person may result more frequently in divorce or break-up due to cultural differences and conflicts in parenting [37]. A study from Finland reported higher divorce rates among native women with a foreign-born partner compared with those with a native partner, which seemed to be explained by unemployment and economic issues in male immigrants [38]. This may partly explain our findings of a higher risk among second-generation immigrants with a foreign-born father and a Swedish-born mother. In addition, cultural differences and conflicts in parenting may also affect these children’s mental health. More data collection is needed, however, to investigate the potential roles of parenting style differences and parental relationships on the mental health of immigrant offspring in Sweden.

The context of the host country and cultural stress may also play major roles in the integration of immigrants and in maintaining their mental health. In a study from the USA, bicultural stress was associated with changes in aggressive and rule-breaking behavior among newly arrived immigrant Hispanic high school adolescents [39]. This population group is also more likely to experience economic difficulties. Immigrants in Sweden have better living conditions compared with those in the UK, Germany, and The Netherlands, as previously reported in a European study [4]. Sweden provides unique social welfare services including unemployment benefits, paid parental leave, monthly child allowance, and free healthcare until the age of 18 to its residents. These services can minimize the impact of economic challenges on lower SES immigrant families. However, poorer social networks and poorer access or utilization of available health care resources may partially explain the lower risks in some vulnerable immigrant groups; if these persons don’t seek medical attention, they won’t be diagnosed. Future research should use other study designs in order to examine whether our findings are true, such as surveys and interviews of parents, teachers and school children. If differences between native and immigrant children/adolescents do exist, the next step would be to examine possible mechanisms behind these potential differences.

### Limitations and Strengths

The results of this study must be interpreted in the light of some limitations. We identified participants with a conduct disorder diagnosis through their inpatient and outpatient hospital referrals in the patient register. While trained doctors with psychiatric expertise have made most of these diagnoses, some inconsistencies in diagnostic procedures over time and across hospitals could not be ruled out. Besides, cultural differences between natives and some immigrants may affect immigrant children’s likelihood of obtaining a conduct disorder diagnosis. Although we accounted for several potential confounders, our analysis did not include all potential confounders (e.g., parental marital status). Future research should attempt to account for residual confounding in their analyses. Despite these limitations, however, this study is the first national cohort study comparing the burden of conduct disorder between immigrant children-adolescents and natives using high quality data. For example, we had no loss to follow up as each person could be tracked for emigration or death because of the unique identification number assigned to each person with a residence permit in Sweden.

## 5. Conclusions

Our results indicated that first-generation immigrants and second-generation immigrants with two foreign-born parents had a lower risk of conduct disorder. In contrast, a markedly higher risk of conduct disorder among male and female second-generation immigrants with a foreign-born father and a Swedish-born mother was found. Future studies should investigate the underlying mechanisms behind these findings including potential inequities in the access to or the utilization of mental health services across natives and different immigrant groups.

## Figures and Tables

**Table 1 ijerph-18-10643-t001:** Characteristics of participants with a native or immigrant background in Sweden, born 1987–2010.

	Natives	First-Generation Immigrants	Second-Generation Immigrants		
	Swedish-Born Parents	Two Foreign-Born Parents	Two Foreign-Born Parents	Foreign-Born Mother ^a^	Foreign-Born Father ^a^
**Included participants, *n* (%)**					
Male	977,897 (51.4)	120,231 (52.5)	144,826 (51.3)	68,608 (51.5)	82,177 (51.1)
Female	924,629 (48.6)	108,658 (47.5)	137,679 (48.7)	64,609 (48.5)	78,662 (48.9)
Total	1,902,526	228,889	282,505	133,217	160,839
**Age at baseline, *n* (%)**					
4–6	1,166,131 (61.3)	66,952 (29.2)	208,306 (73.7)	89,110 (66.9)	104,670 (65.1)
7–11	468,002 (24.6)	80,345 (35.1)	52,060 (18.4)	28,389 (21.3)	36,354 (22.6)
12–16	268,393 (14.1)	81,592 (35.7)	22,139 (7.8)	15,718 (11.8)	19,815 (12.3)
**Maternal income at baseline**					
Quintile 1 (lowest income)	521,609 (27.4)	176,965 (77.3)	183,036 (64.8)	42,221 (31.7)	59,820 (37.2)
Quintile 2	613,257 (32.2)	15,827 (6.9)	55,987 (19.8)	37,742 (28.3)	46,679 (29.0)
Quintile 3	366,150 (19.2)	5907 (2.6)	23,624 (8.4)	24,245 (18.2)	27,783 (17.3)
Quintile 4	233,580 (12.3)	3275 (1.4)	11,136 (3.9)	15,975 (12.0)	15,472 (9.6)
Quintile 5 (highest income)	163,105 (8.6)	3523 (1.5)	5330 (1.9)	11,974 (9.0)	10,109 (6.3)
Missing income	4825 (0.2)	23,392 (10.2)	3392 (1.2)	1060 (0.8)	976 (0.6)
**Had a conduct disorder diagnosis *n* (%)**	6440 (0.4)	454 (0.2)	780 (0.3)	473 (0.4)	941 (0.6)

^a^ Other parent was Swedish-born.

**Table 2 ijerph-18-10643-t002:** Incidence rates of Conduct Disorder (per 100,000 person-years) among participants 4–16 years at baseline in Sweden, 2001–2015.

	Male			Female		
	Person-Years at Risk	Number of Cases	IR (95% CI)	Person-Years at Risk	Number of Cases	IR (95% CI)
**Natives**	7,554,503	4386	58.1 (56.4, 59.8)	7,150,583	2054	28.7 (27.5, 30.0)
**First-generation immigrants**						
Two foreign-born parents	570,751	325	56.9 (51.1, 63.5)	529,726	129	24.3 (20.5, 28.9)
**Second-generation immigrants**						
Two foreign-born parents	1,059,598	540	51.0 (46.8, 55.4)	1,009,784	240	23.8 (20.9, 27.0)
Foreign-born mother ^a^	504,614	313	62.0 (55.5, 69.3)	476,362	160	33.6 (28.8, 39.2)
Foreign-born father ^a^	610,961	639	104.6 (96.8, 113.0)	586,214	302	51.5 (46.0, 57.7)
**Total population**	10,300,427	6203	60.2 (58.7, 61.7)	9,752,969	2885	29.6 (28.5, 30.7)

Conduct Disorder ICD 10 code: F91; Incidence Rate: IR; Confidence Interval: CI. ^a^ Other parent is Swedish-born.

**Table 3 ijerph-18-10643-t003:** Hazard ratios of Conduct Disorder among immigrants 4–16 years at baseline in Sweden, 2001–2015.

	Male		Female	
	Model 1 HR (95% CI)	Model 2 HR (95% CI)	Model 1 HR (95% CI)	Model 2 HR (95% CI)
Natives	Ref.	Ref.	Ref.	Ref.
**First-generation**				
Two foreign-born parents	0.81 (0.72, 0.92)	0.64 (0.57, 0.73)	0.66 (0.54, 0.81)	0.53 (0.43, 0.65)
**Second-generation**				
Two foreign-born parents	0.85 (0.78, 0.93)	0.72 (0.66, 0.79)	0.85 (0.74, 0.97)	0.72 (0.63, 0.83)
Foreign-born mother ^a^	1.05 (0.94, 1.18)	1.04 (0.93, 1.16)	1.19 (1.01, 1.40)	1.18 (1.00, 1.39)
Foreign-born father ^a^	1.79 (1.65, 1.95)	1.71 (1.57, 1.85)	1.82 (1.61, 2.05)	1.73 (1.53, 1.95)

Conduct Disorder ICD 10 code: F91; Hazard ratio: HR; Confidence Interval: CI. Model 1 is adjusted for birth year and age at baseline. Model 2 is also adjusted for maternal income.). ^a^ Other parent is Swedish-born.

## Data Availability

According to the Swedish law, national register data cannot be shared, but it is possible to apply for their use from the authorities.

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
