# Peer review of "Conduct Disorder in Immigrant Children and Adolescents: A Nationwide Cohort Study in Sweden"

_ijerph, 2021, doi:10.3390/ijerph182010643_

Round 1
Reviewer 1 Report
R. 16-18: A clearer (though brief) description of methods is suggested
R. 89 Participants and observation: I suggest a clearer description of the numbers taken into account
R. 117 Immigration background: this section is redoudant
R. Section of references is missing!
Author Response
Dear Editor and Reviewers:
Thank you for your favorable review of our manuscript. We appreciate your many positive comments, e.g., “The work adds to the available literature. I thought the authors did an excellent job reflecting on possible cultural influences on health care use and how these may have influenced their findings." (Reviewer 2); "Osooli M et al. performed a very interesting nationwide retrospective open cohort study with a large data sample on Swedish participants. The data regarding the participants is impressive. The introduction, materials and methods, the results, and discussions (including limitations and strengths) are properly presented, and the study shows promise for guiding future research approaches. That’s why I recommend its publication." (Reviewer 3). We also appreciate your many helpful suggestions for further strengthening the paper. We are now submitting a revised manuscript for your consideration after responding to each of those suggestions. Our detailed responses are listed below.
Reviewer 1
Authors’ reply: We highly appreciate your support and help to improve our manuscript. Please find our specific answers to your queries and comments below.
- 16-18: A clearer (though brief) description of methods is suggested
Authors’ reply: We have clarified several issues in the methods section that needed a more detailed description; this is in accordance with the comments by the reviewers. The current methods section includes, for example, a description of the study design, eligibility criteria, variables, and the data analysis. We could revise the methods section further, if needed.
- 89 Participants and observation: I suggest a clearer description of the numbers taken into account
Authors’ reply: Please see the third sentence in this section: “In total 2,707,976 children-adolescents were included in the study population that consisted of 1,902,526 (70.3%) natives (defined as Swedish-born with both parents born in Sweden), 228,889 (8.4%) first-generation immigrants (defined as born abroad with both parents born abroad), 160,839 (5.9%) second-generation immigrants with a foreign-born father (and a native mother), 133,217 (4.9%) second-generation immigrants with a foreign-born mother (and a native father), and 282,505 (10.4%) second-generation immigrants with two foreign-born parents.”
- 117 Immigration background: this section is redoudant
Authors’ reply: We have deleted this section.
- Section of references is missing!
Authors’ reply: Thank you for noticing! We have added the reference section.
Reviewer 2 Report
This manuscript describes a national cohort study of conduct disorder in children and adolescents in Sweden, focusing in particular on effects of immigration status on the risk of conduct disorder. The work adds to the available literature. I thought the authors did an excellent job reflecting on possible cultural influences on health care use and how these may have influenced their findings, together with the strengths and limitations of a national registry approach to disorder detection.
I do have 3 major concerns about the work and would invite response from the Authors around these points:
1. I understand the rationale for focusing on conduct disorder, but not at excluding attention to other associated mental health disorders (eg, other externalising disorders like ADHD but also internalising disorders like depression and anxiety). Conduct disorder is less prevalent than these other conditions and often co-occurs with them. Presumably other mental health diagnoses could have been / have been extracted from the same national databases?
If the authors do want to keep the focus on conduct disorder specifically, I think they need to make a stronger rationale for this and outline why conduct disorder warrants investigation separately to other mental health difficulties in childhood.
A related point is that some of the Discussion around previous studies of externalising difficulties in childhood (p. 7, para 1) seems to belong in the Introduction.
2. Why was maternal income chosen and not other socioeconomic variables, eg, paternal income, family income, parental education? This is especially relevant given the large group differences in maternal income. It would have been helpful to have other SES variables to consider as well.
3. Follow-up period and age effects: On p. 3, para 1 of the Method it is stated “Approximately 90% of the participants from each of the native/immigrant groups were followed until the end of the study or the age of 18 years.”. What proportion were studied to age 18 vs. just the end of the study? If some children were only studied to, say, age 10 years then they would have fewer years available in which a conduct disorder could have been detected. Please clarify.
Author Response
Dear Editor and Reviewers:
Thank you for your favorable review of our manuscript. We appreciate your many positive comments, e.g., “The work adds to the available literature. I thought the authors did an excellent job reflecting on possible cultural influences on health care use and how these may have influenced their findings." (Reviewer 2); "Osooli M et al. performed a very interesting nationwide retrospective open cohort study with a large data sample on Swedish participants. The data regarding the participants is impressive. The introduction, materials and methods, the results, and discussions (including limitations and strengths) are properly presented, and the study shows promise for guiding future research approaches. That’s why I recommend its publication." (Reviewer 3). We also appreciate your many helpful suggestions for further strengthening the paper. We are now submitting a revised manuscript for your consideration after responding to each of those suggestions. Our detailed responses are listed below.
Reviewer 2
This manuscript describes a national cohort study of conduct disorder in children and adolescents in Sweden, focusing in particular on effects of immigration status on the risk of conduct disorder. The work adds to the available literature. I thought the authors did an excellent job reflecting on possible cultural influences on health care use and how these may have influenced their findings, together with the strengths and limitations of a national registry approach to disorder detection.
I do have 3 major concerns about the work and would invite response from the Authors around these points:
- I understand the rationale for focusing on conduct disorder, but not at excluding attention to other associated mental health disorders (eg, other externalising disorders like ADHD but also internalising disorders like depression and anxiety). Conduct disorder is less prevalent than these other conditions and often co-occurs with them. Presumably other mental health diagnoses could have been / have been extracted from the same national databases?
If the authors do want to keep the focus on conduct disorder specifically, I think they need to make a stronger rationale for this and outline why conduct disorder warrants investigation separately to other mental health difficulties in childhood.
A related point is that some of the Discussion around previous studies of externalising difficulties in childhood (p. 7, para 1) seems to belong in the Introduction.
Authors’ reply: We have recently published studies on ADHD and major depression among immigrants, which also others have done. However, previous research on conduct disorder in immigrant children and adolescents is scarce although it is a diagnosis that should be quite important in the process of integration. We have now explained this in the introduction and also transferred some of the reference to previous literature on immigrant mental health from the discussion to the introduction section. Finally, we now provide reference to our previous research on ADHD and major depression among immigrants in the introduction.
- Why was maternal income chosen and not other socioeconomic variables, eg, paternal income, family income, parental education? This is especially relevant given the large group differences in maternal income. It would have been helpful to have other SES variables to consider as well.
Authors’ reply: We agree that other SES variables could have been considered, such as parental education. However, data on parental education is missing to a high extent among immigrants. The data on income is more reliable as it is based on registered income from the tax authorities, which is the reason why we used maternal income as a surrogate of socioeconomic status. We have clarified this in the methods section:
- Follow-up period and age effects: On p. 3, para 1 of the Method it is stated “Approximately 90% of the participants from each of the native/immigrant groups were followed until the end of the study or the age of 18 years.”. What proportion were studied to age 18 vs. just the end of the study? If some children were only studied to, say, age 10 years then they would have fewer years available in which a conduct disorder could have been detected. Please clarify.
Authors’ reply: Thank you for your relevant query. As stated in the methods section, we have adjusted the regression models for participants’ birth year. In addition, we included a whole national population as our study sample and the distribution of age therefore do not vary substantially across comparison groups.
Reviewer 3 Report
Ossooli M et al. performed a very interesting nationwide retrospective open cohort study with a large data sample on Swedish participants (children and adolescents, aged 4-16 years at baseline) during the study period between Jan 1st, 2001 and Dec 31st, 2015). Their study based on Cox-regression models aimed to compare the conduct disorder risk taken the natives(both parents born in Sweden) as a reference and the first generation immigrant children with two foreign-born parents and second-generation immigrants (foreign-born mother and native father, foreign-born father, and native mother, and two foreign-born subgroups) as predictors adjusted for birth year, age at baseline in the first model and also adjusted for maternal income for the second model. The following categories for the evaluation of time to event in the follow-up were considered: participants that received a diagnosis emigrated, died, turned 18 years, or at the end of the study period, whichever came first. The incidence of conduct disorder and its sub-diagnoses was calculated for each subgroup.
The data regarding the participants is impressive: 2,707,976 participants, of which 1,902,526 natives and 805,450 second-generation immigrants (all subgroups taken together). Data was retrieved from the Swedish national registers and data(RTB, LISA, NPR). To the proposed goal, the authors performed gender-specific models using Cox regression and estimated ratios(HRs) of conduct disorder with 95% CI for the immigrant groups compared with natives. Two models types were generated for each gender(male and female). In Model 1, the authors controlled for birth year and age at the start of the follow up while in Model 2, the authors additionally controlled for maternal income.
The authors properly address a limitation in the current literature related to high-quality studies in immigrant mental health. This limitation refers to most of the studies on immigrant mental health having a suboptimal design such as low sample sizes and problems with bias as a most probable explanation of inconsistent results. The introduction, materials and methods, the results, and discussions (including limitations and strengths) are properly presented, and the study shows promise for guiding futures research approaches. That’s why I recommend its publication. However, I have some comments that the authors should address before the manuscript publication:
Major-the uploaded manuscript lacks the reference list. Some of my comments below could be influenced by this aspect.
Minor:
- Introduction-line 71-75.. When you mention limited data on longitudinal studies(lines 74-76), do you refer to longitudinal studies in Sweden, other countries, or both(current literature)?
- Materials and methods- please either place immigration background subsection(line 117) before participants and observation subsection(line 89) or else in the participants and observation subsection be consistent with defining in brackets also the first generation immigrants like the natives and second-generation immigration subgroups (line 95).
- Results: Is it possible to visually examine the covariates-adjusted survival curves in the respective comparison groups?
- Discussions and conclusions
4.1 Lines 238-248. The authors mention that their study does not have an objective to examine the causal effects of immigration on the risk of conduct disorder, although they attempt to discuss potential explanations behind their findings. Indeed the authors present the risk factors acknowledged in the literature to contribute to developing conduct disorder risk genetic factors, a set of prenatal (e.g., maternal stress, smoking, and drug use), perinatal (e.g., parental psychopathology and malnutrition), familial (e.g., maladaptive and harsh parenting), and neighborhood risk factors (e.g., deviant peers and poverty). However, current literature is scarce in this respect. Hence, the causal pathways and the weight of various risk factors weight in the development of conduct disorder is still a subject for further research. My question is, what further research approaches would the authors suggest based on their findings?
4.2 Line 284 – What does the SES abbreviation stand for?
Author Response
Dear Editor and Reviewers:
Thank you for your favorable review of our manuscript. We appreciate your many positive comments, e.g., “The work adds to the available literature. I thought the authors did an excellent job reflecting on possible cultural influences on health care use and how these may have influenced their findings." (Reviewer 2); "Osooli M et al. performed a very interesting nationwide retrospective open cohort study with a large data sample on Swedish participants. The data regarding the participants is impressive. The introduction, materials and methods, the results, and discussions (including limitations and strengths) are properly presented, and the study shows promise for guiding future research approaches. That’s why I recommend its publication." (Reviewer 3). We also appreciate your many helpful suggestions for further strengthening the paper. We are now submitting a revised manuscript for your consideration after responding to each of those suggestions. Our detailed responses are listed below.
Reviewer 3
Ossooli M et al. performed a very interesting nationwide retrospective open cohort study with a large data sample on Swedish participants (children and adolescents, aged 4-16 years at baseline) during the study period between Jan 1st, 2001 and Dec 31st, 2015). Their study based on Cox-regression models aimed to compare the conduct disorder risk taken the natives(both parents born in Sweden) as a reference and the first generation immigrant children with two foreign-born parents and second-generation immigrants (foreign-born mother and native father, foreign-born father, and native mother, and two foreign-born subgroups) as predictors adjusted for birth year, age at baseline in the first model and also adjusted for maternal income for the second model. The following categories for the evaluation of time to event in the follow-up were considered: participants that received a diagnosis emigrated, died, turned 18 years, or at the end of the study period, whichever came first. The incidence of conduct disorder and its sub-diagnoses was calculated for each subgroup.
The data regarding the participants is impressive: 2,707,976 participants, of which 1,902,526 natives and 805,450 second-generation immigrants (all subgroups taken together). Data was retrieved from the Swedish national registers and data(RTB, LISA, NPR). To the proposed goal, the authors performed gender-specific models using Cox regression and estimated ratios(HRs) of conduct disorder with 95% CI for the immigrant groups compared with natives. Two models types were generated for each gender(male and female). In Model 1, the authors controlled for birth year and age at the start of the follow up while in Model 2, the authors additionally controlled for maternal income.
The authors properly address a limitation in the current literature related to high-quality studies in immigrant mental health. This limitation refers to most of the studies on immigrant mental health having a suboptimal design such as low sample sizes and problems with bias as a most probable explanation of inconsistent results. The introduction, materials and methods, the results, and discussions (including limitations and strengths) are properly presented, and the study shows promise for guiding futures research approaches. That’s why I recommend its publication. However, I have some comments that the authors should address before the manuscript publication:
Major-the uploaded manuscript lacks the reference list. Some of my comments below could be influenced by this aspect.
Authors’ reply: Thank you for your appreciation of our study and valuable comments. Thank you also for noticing that the references were missing. We have now added the reference list.
Minor:
- Introduction-line 71-75.. When you mention limited data on longitudinal studies(lines 74-76), do you refer to longitudinal studies in Sweden, other countries, or both(current literature)?
Authors’ reply: We have added the word “worldwide” to indicate that there is a general lack of longitudinal studies on this topic in the whole world. In addition, we have moved parts of the literature review from the discussion to the introduction. Studies focusing on conduct disorder are very few because of its low incidence and difficulties to collect longitudinal data with adequate statistical power. We have stated this with a reference to a review paper from the Nature Publishing Group in the same paragraph:
“Despite the negative effects of conduct disorder, little is known about the incidence of this disorder across settings and subpopulations (9)”.
- Materials and methods- please either place immigration background subsection(line 117) before participants and observation subsection(line 89) or else in the participants and observation subsection be consistent with defining in brackets also the first generation immigrants like the natives and second-generation immigration subgroups (line 95).
Authors’ reply: The “immigration background” paragraph was considered redundant by one of the reviewers, which we agree with. Therefore, we deleted it.
- Results: Is it possible to visually examine the covariates-adjusted survival curves in the respective comparison groups?
Authors’ reply: It is possible to add graphs presenting adjusted survival curves for the respective comparison groups and we could do it, if requested. However, this information is already presented/summarized in the regression tables.
- Discussions and conclusions
4.1 Lines 238-248. The authors mention that their study does not have an objective to examine the causal effects of immigration on the risk of conduct disorder, although they attempt to discuss potential explanations behind their findings. Indeed the authors present the risk factors acknowledged in the literature to contribute to developing conduct disorder risk genetic factors, a set of prenatal (e.g., maternal stress, smoking, and drug use), perinatal (e.g., parental psychopathology and malnutrition), familial (e.g., maladaptive and harsh parenting), and neighborhood risk factors (e.g., deviant peers and poverty). However, current literature is scarce in this respect. Hence, the causal pathways and the weight of various risk factors weight in the development of conduct disorder is still a subject for further research. My question is, what further research approaches would the authors suggest based on their findings?
Authors’ reply: Thank you for this important comment. We have elaborated more specifically on future research approaches such as in the statement below.
“More data collection is needed, however, to investigate the potential roles of parenting style differences and parental relationships on the mental health of immigrant offspring in Sweden.”
“However, poorer social networks and poorer access or utilization of available health care resources may partially explain the lower risks in some vulnerable immigrant groups; if these persons don’t seek medical attention, they won’t be diagnosed.”
We also added the following sentences to the end of the statements above as a future research direction:
“Future research should use other study designs in order to examine whether our findings are true, such as surveys and interviews of parents, teachers and school children. If differences between native and immigrant children/adolescents do exist, the next step would be to examine possible mechanisms behind these potential differences.”
4.2 Line 284 – What does the SES abbreviation stand for?
Authors’ reply: We replaced the abbreviation with the full word, i.e., “socioeconomic status”.
Reviewer 4 Report
- The abstract exceeds the permitted word count of 200 words by almost 50%
- The abstract as well as the introduction fail to inform the reader about the hypothesis and the benefit of the conducted study. In the beginning of the introduction, the authors list general factors which may increase the risk to develop mental health problems in people with immigration background. However, these risk factors are neither specific to children nor to conduct disorder. Furthermore, the authors do not discuss the current literature on this topic. The literature which is mentioned in the discussion, should be mentioned and discussed in the introduction in order to derive hypotheses. Thus, it currently remains unclear a) why the authors suspect that native children compared to children with various immigration backgrounds differ in their risk to develop CD b) how the field could benefit from this research.
- The authors do not explain why maternal income was considered as a covariate. Why maternal income and not socio economic status of the family, which is a known risk factor?
- The authors seem to mix up ICD 10 and DSM -5 conduct disorder diagnosis in their methods section as well as in the tables of the appendix. F91.1 is not childhood-onset type but unsocialized conduct disorder, F91.2 is not adolescent-onset type but socialized conduct disorder.
- There seems to be an error in the formatting of table 1, as the first three lines (included participants) of table 1 page 6 are repeated on page 5.
- I am not an expert when it comes to cox regression but I believe that the results section would benefit from more detail (e.g. include the p-values, which post-hoc testss were used to determine significant differences between individual groups, were the other predictors (age at baseline, birth year, maternal income) significant as well?)
- In their discussion the authors list a couple of arguments why children with immigration background may have a higher risk to show more health problems. However, they found that most of the invetsigated immigration groups had a lower risk. Thus, the authors fail to explain their findings.
Author Response
Dear Editor and Reviewers:
Thank you for your favorable review of our manuscript. We appreciate your many positive comments, e.g., “The work adds to the available literature. I thought the authors did an excellent job reflecting on possible cultural influences on health care use and how these may have influenced their findings." (Reviewer 2); "Osooli M et al. performed a very interesting nationwide retrospective open cohort study with a large data sample on Swedish participants. The data regarding the participants is impressive. The introduction, materials and methods, the results, and discussions (including limitations and strengths) are properly presented, and the study shows promise for guiding future research approaches. That’s why I recommend its publication." (Reviewer 3). We also appreciate your many helpful suggestions for further strengthening the paper. We are now submitting a revised manuscript for your consideration after responding to each of those suggestions. Our detailed responses are listed below.
Reviewer 4
- The abstract exceeds the permitted word count of 200 words by almost 50%
Authors’ reply: We have reduced the word count of the abstract from 261 to 200 words.
- The abstract as well as the introduction fail to inform the reader about the hypothesis and the benefit of the conducted study. In the beginning of the introduction, the authors list general factors which may increase the risk to develop mental health problems in people with immigration background. However, these risk factors are neither specific to children nor to conduct disorder. Furthermore, the authors do not discuss the current literature on this topic. The literature which is mentioned in the discussion, should be mentioned and discussed in the introduction in order to derive hypotheses. Thus, it currently remains unclear a) why the authors suspect that native children compared to children with various immigration backgrounds differ in their risk to develop CD b) how the field could benefit from this research.
Authors’ reply: We have transferred parts of the literature review from the discussion to the introduction section. We have also added new text to explain why immigrant children may have a higher risk of poor mental health and why it is important to examine conduct disorders in immigrant children and adolescents:
“Despite the negative effects of conduct disorder, little is known about the incidence across settings and subpopulations (9) and more knowledge of this disorder is particularly important in immigrants as the many negative effects of this disorder could hinder integration in the new society. It is also possible that the psychosocial stress related to migration may influence children’s and adolescents’ mental health.”
“Identifying subpopulations at higher risks of conduct disorder is a crucial step in planning and delivering prevention and treatment interventions.”
- The authors do not explain why maternal income was considered as a covariate. Why maternal income and not socio economic status of the family, which is a known risk factor?
Authors’ reply: We considered using other variables to determine socioeconomic status, such as parental education. However, data on parental education is missing to a high extent among immigrants. The data on income is more reliable as it is based on registered income from the tax authorities, which is the reason why we used maternal income as a surrogate of socioeconomic status. Maternal income is strongly related to family socioeconomic status, according to the literature that we have reviewed.
- The authors seem to mix up ICD 10 and DSM -5 conduct disorder diagnosis in their methods section as well as in the tables of the appendix. F91.1 is not childhood-onset type but unsocialized conduct disorder, F91.2 is not adolescent-onset type but socialized conduct disorder.
Authors’ reply: Thank you for pointing out this error. We have revised both the methods and the appendix table headings to correct this.
- There seems to be an error in the formatting of table 1, as the first three lines (included participants) of table 1 page 6 are repeated on page 5.
Authors’ reply: We appreciate this comment and have fixed this.
- I am not an expert when it comes to cox regression but I believe that the results section would benefit from more detail (e.g. include the p-values, which post-hoc testss were used to determine significant differences between individual groups, were the other predictors (age at baseline, birth year, maternal income) significant as well?)
Authors’ reply: Thank you for this comment. We could add p-values but feel that this would cause space constraints. Confidence intervals are also useful to indicate statistical significance and are often used instead of p-values. Yes, we observed lower risks of conduct disorder among more recent birth cohorts and among those with higher maternal income at baseline. However, age at baseline was not associated with risk of conduct disorder at a statistically significant level. We did not present the significance levels of the potential confounders in order to have a stronger focus on the main research question.
- In their discussion the authors list a couple of arguments why children with immigration background may have a higher risk to show more health problems. However, they found that most of the invetsigated immigration groups had a lower risk. Thus, the authors fail to explain their findings.
Authors’ reply: Thank you for this comment. We were unable to investigate the causality behind our findings but have tried to elaborate on the potential explanations behind our findings. Please see the last two paragraphs of the discussion where we have attempted to provide some explanations behind our findings.
We also added the following sentences in order to suggest future research directions:
“Future research should use other study designs in order to examine whether our findings are true, such as surveys and interviews of parents, teachers and school children. If differences between native and immigrant children/adolescents do exist, the next step would be to examine possible mechanisms behind these potential differences.”
Round 2
Reviewer 2 Report
The authors have revised the manuscript comprehensively and I have no further suggestion for change.
Author Response
We thank you for your invaluable support and helping us to improve our work.
Reviewer 4 Report
Most of the made comments were adequately answered by the authors and the introduction and methods grately improved.
I completely agree with the authors that they cannot make any causal conclusions based on their analysis. However, one should always consider possible explanations for study results as they are extremly valuable for new research. Considering that the authors initially list reasons why children with immigration background may have an increased risk to develop CD due to higher mental health burdens etc., I would expect a discussion of possible reasons why they (and seemingly most of the cited literature) found the opposite to be true in children with two foreign born parents. For example, is it possible that the mentioned difference in usage of healthcare services in different immigration populations, may lead to an underdiagnosis of CD in children of two immigrant parents? Thus, is it possible that the results do not reflect a decreased risk but rather a higher number of undetected cases due to a lower likelyhood of seeking the help of a mental healthcare professional?
The limitations should also mention other possibly influencing factors that the study did not include (e.g. marietal status of the parents) and which future research should consider.
Author Response
Dear Reviewer,
Thank you so much for reviewing our revised manuscript. We appreciate your helpful comments and your interest in helping us to improve our work further. We have addressed both of your queries.
Most of the made comments were adequately answered by the authors and the introduction and methods grately improved.
I completely agree with the authors that they cannot make any causal conclusions based on their analysis. However, one should always consider possible explanations for study results as they are extremly valuable for new research. Considering that the authors initially list reasons why children with immigration background may have an increased risk to develop CD due to higher mental health burdens etc., I would expect a discussion of possible reasons why they (and seemingly most of the cited literature) found the opposite to be true in children with two foreign born parents. For example, is it possible that the mentioned difference in usage of healthcare services in different immigration populations, may lead to an underdiagnosis of CD in children of two immigrant parents? Thus, is it possible that the results do not reflect a decreased risk but rather a higher number of undetected cases due to a lower likelyhood of seeking the help of a mental healthcare professional?
Authors’ reply: Thank you for a very relevant and important comment. In the fifth paragraph of the discussion, we have discussed the possible lower healthcare service utilization of psychiatric disorders among immigrants as a potential explanation for our findings.
“Among all immigrant groups, only those second-generation immigrants with a foreign-born father and a Swedish-born mother had markedly higher risks of conduct disorder compared with natives while first-generation immigrant children and adolescents as well as second-generation immigrants with two foreign-born parents had lower risks. All residents in Sweden are entitled to free healthcare until they turn 18 years. Moreover, schools are actively engaged in maintaining the mental health of their students, but the use of these services may be hampered because some immigrant populations have different patterns of using healthcare services for their children’s psychiatric problems (33, 34). Poor health literacy, cultural barriers, and stigma related to psychiatric disorders are more common among immigrant parents than natives and especially in those with a low socioeconomic status (35, 36). On the other hand, because conduct disorder involves the violation of other peoples’ rights, lower service utilization may play a less important role in its diagnosis compared with other psychiatric disorders. Raising awareness on early symptoms of conduct disorder among parents, guardians, and teachers can help to ensure early diagnosis and start of psychosocial support among affected children and adolescents.”
In addition, we have also highlighted potential differences in obtaining a diagnosis between immigrants and natives in the “limitations and strengths” section.
“Besides, cultural differences between natives and some immigrants may affect immigrant children’s likelihood of obtaining a conduct disorder diagnosis.”
Finally, we have mentioned potential differences in healthcare service access and utilization as a potential reason for our observed findings.
“Future studies should investigate the underlying mechanisms behind these findings including potential inequities in the access to or the utilization of mental health services across natives and different immigrant groups.”
The limitations should also mention other possibly influencing factors that the study did not include (e.g. marietal status of the parents) and which future research should consider.
Authors’ reply: We added the following, new limitation as well.
“Although we accounted for several potential confounders, our analysis did not include all potential confounders (e.g., parental marital status). Future research should attempt to account for residual confounding in their analyses.”